# Epigenome-Wide Analysis Reveals DNA Methylation Alteration in *ZFP57* and Its Target *RASGFR2* in a Mexican Population Cohort with Autism

**DOI:** 10.3390/children9040462

**Published:** 2022-03-25

**Authors:** Queletzu Aspra, Brenda Cabrera-Mendoza, Mirna Edith Morales-Marín, Carla Márquez, Carlos Chicalote, Ana Ballesteros, Miriam Aguilar, Xochitl Castro, Amalia Gómez-Cotero, Ana María Balboa-Verduzco, Lilia Albores-Gallo, Omar Nafate-López, Carlos Alfonso Marcín-Salazar, Patricia Sánchez, Nuria Lanzagorta-Piñol, Fernando Omar López-Armenta, Humberto Nicolini

**Affiliations:** 1Genomics of Psychiatric and Neurodegenerative Diseases Laboratory, National Institute of Genomic Medicine (INMEGEN), Mexico City 14610, Mexico; queletzu@gmail.com (Q.A.); memorales@inmegen.gob.mx (M.E.M.-M.); carlitajam@gmail.com (C.M.); aguilarmiriam861@gmail.com (M.A.); xcastro@inmegen.gob.mx (X.C.); 2Biological Sciences Graduate School, National Autonomous University of Mexico, Mexico City 04510, Mexico; 3PECEM, Faculty of Medicine, National Autonomous University of Mexico, Mexico City 04510, Mexico; brenda.cabrera-mendoza@yale.edu; 4Centro de Investigación y Atención en Salud Mental Kaelum Neurocenter S.C., Mexico City 53100, Mexico; drchicalote@gmail.com; 5Hospital Central PEMEX, Mexico City 02720, Mexico; teresaballesterosm@gmail.com; 6Centro Interdisciplinario de Ciencias de la Salud (CICS-UST) Unidad Santo Tomás, Instituto Politécnico Nacional, Mexico City 11340, Mexico; amaliagpegomez@prodigy.net.mx (A.G.-C.); ana_maria_balboa@hotmail.com (A.M.B.-V.); 7Hospital Psiquiátrico Infantil Juan N. Navarro, Secretaría de Salud, Mexico City 14080, Mexico; lilialbores@gmail.com; 8Unidad de Neuropsiquiatría, Hospital de Especialidades Pediátricas de Tuxtla Gutiérrez, Tuxtla Gutiérrez 29070, Mexico; axlnafate@hotmail.com; 9Clinical Researcher at Clínica Mexicana de Autismo (CLIMA), Mexico City 03740, Mexico; psycho_coorp@yahoo.com (C.A.M.-S.); patricia.s.lizardi@gmail.com (P.S.); 10Carracci Medical Group, Mexico City 03740, Mexico; lanzagorta_nuria@gmc.org.mx (N.L.-P.); lopezarmenta.fdl@gmail.com (F.O.L.-A.)

**Keywords:** Autism, DNA methylation, imprinting, children, Mexican

## Abstract

Autism Spectrum Disorders (ASD) comprise a group of heterogeneous and complex neurodevelopmental disorders. Genetic and environmental factors contribute to ASD etiology. DNA methylation is particularly relevant for ASD due to its mediating role in the complex interaction between genotype and environment and has been implicated in ASD pathophysiology. The lack of diversity in DNA methylation studies in ASD individuals is remarkable. Since genetic and environmental factors are likely to vary across populations, the study of underrepresented populations is necessary to understand the molecular alterations involved in ASD and the risk factors underlying these changes. This study explored genome-wide differences in DNA methylation patterns in buccal epithelium cells between Mexican ASD patients (*n* = 27) and age-matched typically developing (TD: *n* = 15) children. DNA methylation profiles were evaluated with the Illumina 450k array. We evaluated the interaction between sex and ASD and found a differentially methylated region (DMR) over the 5′UTR region of *ZFP57* and one of its targets, *RASGRF2.* These results match previous findings in brain tissue, which may indicate that *ZFP57* could be used as a proxy for DNA methylation in different tissues. This is the first study performed in a Mexican, and subsequently, Latin American, population that evaluates DNA methylation in ASD patients.

## 1. Introduction

Autism Spectrum Disorders (ASD) comprise a group of complex neurodevelopmental disorders, defined by an abnormal development of restricted interests, stereotypical behavior, limited social interaction and communication according to the 5th edition of the Diagnostic and Statistical Manual of Mental Disorders (DSM-5) [1]. The global prevalence for ASD is about one in 160 born children [2,3]. This prevalence has a strong male bias, as the ASD male-female ratio is 2.6 to 5.2 boys per one girl [4]. Among possible mechanisms contributing to this high male-female ratio are the female protective effect (females have a higher threshold on environmental and/or genetic risk for reaching the same affection status than males), sex-differential genetic and hormonal factors [5,6,7].

Several factors contributing to ASD etiology have been identified, including genetic and environmental factors [8,9]. Regarding genetic factors, there is strong evidence of their key role in the vulnerability to ASD, as the heritability estimates from twin studies are as high as 90% [10]. Some important genetic factors identified in ASD are those related to specific biological pathways, including synaptogenesis and synaptic function [11,12,13], such as the *CAPG* gene that encodes for a postsynaptic density protein [14]. On the other hand, among environmental factors associated with ASD are parental age, zinc deficiency, and maternal physical and mental health [15,16]. Environmental factors may exert their influence in the ASD phenotype through epigenetic modifications, such as DNA methylation [17]. DNA methylation and other epigenetic changes are particularly relevant for ASD due to their mediating role in the complex interaction between genotype and environment [9].

Several DNA methylation alterations have been identified in the brain tissue of individuals with ASD and have been implicated in their pathophysiology [9,18,19,20,21]. Although postmortem brain tissue studies have provided relevant insight into ASD neurobiology, the access to brain tissue samples is limited. In this context, the study of DNA methylation changes in peripheral tissues in ASD might contribute to a wider understanding of the DNA methylation changes associated with ASD. Similarities in DNA methylation patterns in saliva, brain, and blood have been found in patients with schizophrenia [22] and healthy individuals [23]. Thus, peripheral DNA methylation changes that reflect alterations found in brain tissue are particularly relevant, as they could be used as proxies of brain DNA methylation changes in easily accessible samples in living patients [24].

The search for peripheral DNA methylation changes in ASD is a largely unexplored research area as only a few studies have been conducted. In blood, seven regions with concordant DNA methylation levels in the prefrontal cortex, temporal cortex, and cerebellum have been identified [25]. In addition, DNA methylation differences were evaluated in buccal epithelial cells between individuals with ASD born to mothers of >35 years and typically developing controls. This study detected 13 differentially methylated regions (DMRs), mainly over genes with copy number variant (CNV) reports between ASD and typically developing controls [26].

The lack of diversity in DNA methylation studies in ASD individuals is remarkable. Genetic and environmental factors are likely to vary across populations, i.e., several population specific variants that increase ASD susceptibility have been identified in non-caucasian populations [27]. The aim of our study was to identify ASD-associated changes in DNA methylation patterns in buccal epithelium from Mexican individuals with ASD. To our knowledge, this is the first genome-wide DNA methylation study performed in individuals with ASD from a Latin American population.

## 2. Materials and Methods

### 2.1. Subjects and Samples

Participant recruitment and sample collection was performed with the approval of the Ethics Committee of the Carracci Medical Group, in Mexico City, Mexico. The present study was performed in accordance with the Helsinki Declaration protocol code: AUTs/n-1 and date of approval 11 October 2010. All subjects or their legal representatives in the case of minors provided their written informed consent. In addition, minors provide their assent to participate in the present study.

We enrolled individuals with diagnosed ASD (*n* = 27), and a control group of typically developing (TD) individuals (*n* = 15). We recruited participants with ASD from: Hospital Psiquiátrico Infantil Juan N. Navarro, Centro de Investigación y Atención en Salud Mental Kaelum Neurocenter S.C., Mexico City, Clínica Mexicana de Autismo (CLIMA) and Apapache Autismo Apoyo para padres con hijos en espectro Autista (APAPACHE) non-profit association in Mexico City. TD children were contacted via flyers distributed in daycare centers and preschool. ASD and TD children were Mexicans. Every individual in the ASD group had a pre-existing diagnosis of ASD, which was confirmed with a parent interview (Autism Diagnostic Interview-Revised—ADI-R) [28]; and clinical diagnosis based on DSM-5 [1] criteria by a child psychiatrist as detailed in Marquéz et al., 2019 [29].

The ADI-R [28] is a parent structured, standardized interview to collect developmental information of current and historical presentation of their children with autistic symptoms. The interview is composed of relevant items that refer to the core domains of ASD covering the following subdimensions: (i) Communication and language impairments, (ii) Impairments in social interactions, (iii) Repetitive and restricted behaviors. The Clinical variables of ADI-R are scored on a 0 to 3 scale, where 0 indicates normal response and 3 indicates the highest level of severity. Responses are recorded separately for current behavior and past behavior, as there might be some cases where the symptoms disappeared, but diagnosis should be made if behaviors occurred in the past. We considered the following cutoff points for each subdomain: Communication and language impairment ≥8; Impairment in social interactions ≥10; Repetitive and restricted behaviors ≥3. All participants with ASD met the cutoff points considered. Evaluators were required to complete a training course in order to carry out the interview.

Participants of this study were selected randomly from a previously described sample [29,30]. We selected 27 patients with a psychiatric diagnosis of ASD with ages ranging from 3 to 12 (mean = 5.2, SD= 1.9). Our typically developing group (TD) was composed of 15 children with ages ranging from 4 to 7 (mean = 5.6, SD = 1.09) without ASD diagnosis and evaluated through the Social Responsiveness Scale to discard the presence of autistic symptomatology and confirm their eligibility as TD individuals for this study [31,32].The Social Responsiveness Scale (SRS) is a quantitative measure of autistic traits, which has been used in behavior-genetic, epidemiological and intervention studies. It has cross-cultural validity and analyses yield single factors for the normative and clinical samples [33] Samples from buccal mucosa using swabs were obtained during the evaluation. Sex differences among the ASD and TD groups were assessed with a Fisher’s exact test. Meanwhile, age differences between both groups were evaluated by a Kruskal–Wallis chi-squared test. The characteristics of the ASD and TD groups are summarized in Table 1.

### 2.2. DNA Extraction and Bisulfite Conversion

Genomic DNA from the buccal epithelial samples was isolated following the protocol for Gentra Puregene Blood Kit from Qiagen. DNA concentration and integrity were determined using the NanoDrop 2000 spectrometer (Thermo Fisher, Wilmington, DE, USA) and agarose gel electrophoresis, respectively. For genotyping assessment, 500 ng of genomic DNA was hybridized into the Infinium PsychArray BeadChip (Illumina, San Diego, CA, USA).

Then, for DNA methylation profile assessment, 500 ng of the isolated DNA was bisulphite-conversed using the EZ 96-DNA methylation kit (Zymo Research, Irvine, CA, USA) following the manufacturer’s standard protocol. Genome-wide DNA methylation profile was assessed using the Illumina Infinium HumanMethylation450 Beadchip platform (Illumina, San Diego, CA, USA). The processing of both arrays was performed according to the recommendations of the manufacturers and protocols and were scanned on an iScan Microarray Scanner for microarray signal detection (Illumina, San Diego, CA, USA) immediately after their respective protocols.

### 2.3. Ancestry Analysis

We used PLINK 1.9 [34] software to perform quality control of the genotyping data following the recommendations of Anderson et al. [35]. SNPs were removed if they did not accomplish the following criteria: (i) a *p*-value lower than 1 × 10^−6^ for a Hardy–Weinberger equilibrium (HWE) test, (ii) a SNP minor allele frequency (MAF) higher than 0.05, (iii) a SNP call rate lower than 95%. Participants were excluded if they had a genotyping call rate lower than 95% or exhibited relatedness to other individuals in the sample (PI_HAT ≥ 0.5).

An ancestry principal component analysis (PCA) was performed with PLINK as shown in Figure 1 [27]. To include only linkage disequilibrium (LD)-independent SNPs in the ancestry principal components analysis, we performed a LD pruning, considering a window size of 50 kb, a variant count to shift the window at the end of each step of 5, and a pair pairwise correlation threshold of 0.2. Ancestry-informative markers were obtained from Hapmap phase 3 data (HapMapIII) [36] using the following populations as references: Utah Residents with Northern and West European ancestry (CEU), Han Chinese in Beijing, China (CHB), Yoruba in Ibadan, Nigeria (YRI), and Japanese in Tokyo, Japan (JPT) to trace a PCA analysis with PLINK software to determine if the participants of this study were evenly distributed between CEU and CHB populations (Figure 1).

### 2.4. Genome-Wide Analysis of Differential DNA Methylation

We conducted DNA methylation data preprocessing using the minfi R package [37]. First, probes were filtered using the following criteria: (i) mean detection *p*-value > 0.01; (ii) probes previously reported as cross-reactive [38] (iii) control probes and (iv) probes located in the X and Y chromosomes. After filtering, we obtained 426544 probes. Then, data were normalized using the quantile method [39].

DNA methylation differences between ASD patients and TD individuals were evaluated with a linear model implemented in limma including sex as a covariate [40]. Considering males are four times more likely to have ASD than females [41], the existing functional evidence of an interaction between sex and ASD diagnosis [34] and the recent findings of sex differences in DNA methylation in idiopathic autism [42]; we included sex as an interacting variable in our comparisons. *p*-values associated with differentially methylated sites between the groups were adjusted for a multiple comparisons correction using the Benjamini—Hochberg correction for a false discovery rate (FDR) [43]. An FDR corrected *p*-value < 0.05 was considered statistically significant in this study.

Then, we used DMRcate (Bioconductor) [44] to find differentially methylated regions between ASD and TD. We included regions that had more than four CpGs with an FDR corrected *p*-value < 0.05 and at least a mean beta difference of 5%. To find differentially methylated regions between ASD and TD children, considering sex as an interactive variable, we applied more astringent settings in order to avoid false positives. We included only those with more than eight CpGs with an FDR corrected *p*-value < 0.05 and at least a mean beta difference of 10%. These methods are equivalent to those used in previous studies [20].

Gene regions were derived from the human assembly GRCh37 in the Ensembl genome browser via biomaRt and the genes and structural variations found in each region were identified [45,46]. All the analyses were performed in the R environment [47].

## 3. Results

### 3.1. Clinical and Demographic Data

Comparison between groups showed no significant differences in the ages of children (mean ASD group= 5.2, SD = 1.9, mean reference group = 5.06, SD = 1.09, Kruskal–Wallis chi-squared = 4.132, df = 7, *p*-value = 0.7645). On the other hand, a Fisher’s exact test revealed sex differences between the groups (*p*-value =0.01479). This is consistent with previous reports indicating that males are more frequently affected by ASD than females [4].

### 3.2. Differentially Methylated Regions between ASD and TD Individuals

We performed a genome-wide DNA methylation screen on buccal epithelium and identified a total of four significant (FDR corrected *p*-value < 0.05) DMRs between ASD patients and TD individuals. These four DMRs were located in the vicinity of genes FAIM2, Fas Apoptotic Inhibitory Molecule 2; CPXM2, Carboxypeptidase X, M14 Family Member 2; NRIP2, Nuclear Receptor Interacting Protein 2; SOX7, SRY-Box 7 and CTD-2135J3.3 as shown in Table 2.

### 3.3. Differentially Methylated Regions Affected by Sex and ASD Diagnosis

We found two DMRs that result from the interaction between ASD and sex (FDR corrected *p*-value < 0.05, CpGs > 8, Methylation differences (β, ASD-TD) > 0.1). As shown in Figure 2, we identified the first DMR over the 5′UTR region of *ZFP57*, encoding for zinc finger protein 57. This DMR consists of 24 differentially methylated probes with an average of a 0.1 mean beta difference. This region extends through 931 base pairs with hypermethylation in the tissue of autistic individuals (ASD Females shown in red, ASD Males shown in blue, TD Females shown in green, TD Males shown in purple) compared with TD individuals. On average, females and males with ASD have 10% higher methylation levels than TD females and males at this DMR (FDR corrected *p*-value = 0.0038). We did not find any significant differences between ASD and TD individuals regarding CNV status with this DMR as shown in Figure 2. We also found one of *ZFP57* gene targets *RASGRF2* [37] in a state of hypomethylation in ASD individuals (FDR corrected *p*-value for the DMR = 0.00033, mean beta fold change among the six CpG sites of this region = −4.036155 × 10^−2^, mean FDR corrected *p*-value among the six CpG sites of this region = 2.156111 × 10^−6^). This hypomethylated region spans through 468 base pairs around six CpG sites and it is located over the edge of a CpG Island, and its shore is on the first exon and part of the gene body of *RASGRF2*. The hypomethylated region in *RASGRF2* is shorter than eight CpGs, which was our threshold to define a DMR. Thus, it was not detected in our main analysis. The hypomethylation of *RASGRF2* was found after a targeted search of the methylation status of *ZFP57* targets due to its potential relevance in ASD.

The second DMR expands over 839 base pairs that overlap the 5′UTR region of GSTT1, Glutathione S-Transferase Theta 1. This DMR comprises 12 differentially methylated probes that are on average 10% less methylated in ASD individuals compared to TD subjects (FDR corrected *p*-value = 0.0094).

Females of both groups tend to have higher methylation levels than males. This region spans throughout a CpG Island and its flanking shores, from the 200 base pair upstream the transcription start site through the first exon of gene GSTT1. We did not find any difference regarding CNV status along this DMR between ASD and TD individuals as shown in Figure 3.

## 4. Discussion

Our study revealed different DNA methylation patterns between individuals with ASD and TD children in a Mexican sample. In Mexico the ASD prevalence reported is higher than the global one (1:116) [3]. Therefore, the identification of ASD risk factors in the Mexican population is crucial to design preventive strategies for ASD and other neurodevelopmental disorders in this population. Thus, the inclusion of underrepresented populations, i.e., Mexican and other Latin American populations, is essential for the identification of population-specific molecular alterations and risk factors in ASD.

The oral cavity has shown to be a great source of easily accessible biological material such as DNA, by its quick and low-cost collection compared to other tissues such as blood [48]. Furthermore, obtaining samples from buccal mucosa is a non-invasive procedure in contrast to other peripheral tissues, i.e., blood [49]. This is highly relevant and a potential advantage when evaluating ASD patients, particularly those at a pediatric age, where invasive medical procedures might be triggers for anxiety and stress [50].

Since epithelial cells are derived from ectoderm, they are likely to reflect nervous system changes [25,48,51]. Indeed, DNA methylation from saliva has shown to be more similar to that of brain tissues compared with DNA methylation from blood [25]. The only previous report of DNA methylation alterations related with ASD in buccal epithelial cells was conducted comparing individuals with ASD born to mothers of >35 years and typically developing controls [27]. The DNA methylation changes found in this previous study and ours suggest, like previous authors have, that early embryological events alter epigenomic landscape in ectoderm derived tissues such as the central nervous system and epithelial cells [27].

### 4.1. Differentially Methylated Regions between ASD and TD Children Relate to Genes Involved in Neurodevelopment

The present work found four DMRs between individuals with ASD and TD children that have not been previously associated with ASD that can suggest future investigation paths. The first one is located in *FAIM2*, which according to the HBT (Human Brain Transcriptome) database [52,53] is expressed increasingly after birth [54]. Knocking out the expression of *FAIM2* in rats produces abnormal cell morphology and reduced cell density in Purkinje cells in the cerebellum [55], which resembles similar alterations in ASD [56]. In the same sense, *CPXM2* is expressed in the mediodorsal nucleus of the thalamus, and it has been detected in the macrophage/migroglia in the brains of multiple sclerosis patients [57]; this finding corresponds with recent evidence of the role of microglia in the development of normal neural networks and ASD [58]. Furthermore, SOX7 protein through its interaction with β-catenin has been described to promote cerebellar neuronal apoptosis [59]. SOX7 represses Wnt/β-catenin and is expressed during development in brain, heart, lung, kidney, colon, spleen, and epithelio-mesenchymal transitions [60]. On the same note NRIP2, Nuclear Receptor Interacting Protein 2, participates in the upregulation of the Wnt pathway, which plays a key role in neurodevelopment [61]. Our results concur with previous studies performed in Hispanic and non Hispanic white subjects with ASD in contrast with TD subjects, although we did not find alterations in DNA methylation in the same genes, we found altered methylation in genes related to neurodevelopment [42].

### 4.2. Differentially Methylated Regions in ASD Interact with Sex

Therefore, we identified two DMRs after considering sex as an interaction variable. The first DMR resides over the 5′UTR of *ZFP57.* Of note, this region has been previously reported as a DMR in postmortem brain tissue of ASD patients from the Harvard Brain Tissue Resource Center [19]. We found the same region as reported, though the region we identified extends 200 base pairs upstream from the coding region of *ZFP57*. It is important to note that even though this previous report has fewer female ASD individuals, the pattern of methylation we found is the same: ASD individuals tend to have higher methylation levels regarding TD individuals and females tend to have the lowest values in both groups. The relevance of this finding is remarkable, as we found in peripheral tissue a region that potentially could be a proxy of methylation status in the brain. Future studies are needed to confirm these results in other samples.

Furthermore, we found one of the *ZFP57* targets: *RASGRF2* to be hypomethylated in ASD individuals. The adequate expression of ZFP57 maintains DNA methylation in imprinting control regions and in turn affects allele-specific gene expression [62]. Parent of origin alleles and imprinting have been related to the development of ASD [63]. Even though we used a surrogate cell type, we found differences in regions that had been previously reported in brain tissue. These findings imply that we may have found a novel region representing DNA methylation patterns across tissue in ASD [23,24].

It is important to add that methylation levels in *ZFP57* can be altered due to low folate intake during pregnancy, resulting in hypermethylation in the same region we report [64]. This becomes vitally important in the Mexican population, as the polymorphism C677T *MTHFR* (methylenetetrahydrofolate reductase) has different allele frequencies throughout the Mexican territory [65] and has an effect on folate metabolism [66]. Folate deficiency is a well-known factor involved in neurodevelopment alterations. The C677T variant has shown to be correlated with a higher risk of developing cleft palate in offspring [67]. Maternal intake of dietary methyl-group donors during the periconceptional period is associated with alterations in DNA methylation of genes related to growth [67]. In the same sense, maternal socioeconomic status has been shown to be an indicator of DNA methylation of LINE-1 and Alu repeat elements in a Mexican-American cohort [68].

The second DMR is located near *GSTT1* (Glutathione S-Transferase Theta 1). The null genotype of *GSTT1* has been reported to convey higher ASD risk odds in a Jamaican cohort [69]. The null genotype for *GSTT1* has also been associated with potential liver damage due to valproic acid intake, which is common in ASD patients [70]. Even if we did not assess the absence of this genotype, our estimation of copy number variants (Figure 3, lower panel) did not indicate differences between patients and controls.

Future research needs to determine the impact of DNA methylation in the expression of *GSST1* and its effect over valproic acid metabolism.

Limitations of the present study must be acknowledged. First, the relatively small sample. However, the differences we found let us make inferences that relate to previous findings [14]. The amount of methylation change we identified ranged from 5.2–18.2%, which allows us to compare it to previous studies [14,27]. Therefore, future research with larger samples and individuals of both sexes is necessary to confirm our results and their association with ASD. Second, the restriction of this study to Mexican subjects. The environmental factors to which subjects in the Mexican population are exposed may lead to specific epigenetic changes not found in other populations.

## 5. Conclusions

We identified DMRs between individuals with ASD and TD from a Mexican population. The interaction between sex and ASD was associated with DNA methylation changes in the 5′UTR region of *ZFP57*, and one of its targets *RASGRF2*. This finding matched previous reports on brain tissue, which may indicate that *ZFP57* could be used as a proxy of methylation in different tissues. The further evaluation of population-specific DNA methylation alterations will contribute to the further identification of genetic risk and environmental factors involved in ASD, as well as to the design of strategies for their prevention.

## Figures and Tables

**Figure 1 children-09-00462-f001:**
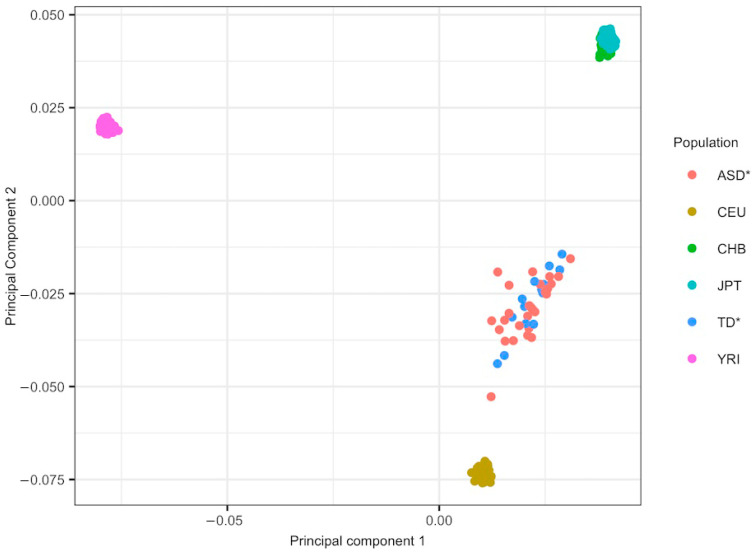
Principal components analysis (PCA) of participants’ genotypes along with HapMap3 populations data. CEU: Utah Residents with Northern and West European ancestry; CHB: Han Chinese in Beijing, China; YRI: Yoruba in Ibadan, Nigeria; JPT: Japanese in Tokyo, Japan; ASD*: Autism Spectrum Disorder diagnosed subjects from this study; TD*: Typically developing subjects from this study.

**Figure 2 children-09-00462-f002:**
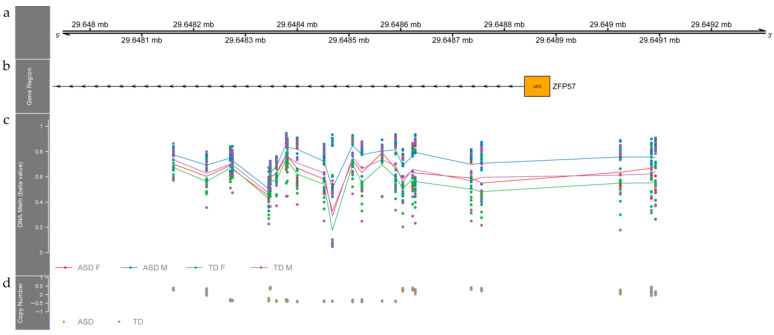
A genome-wide differentially methylated region (DMR) located in the 5′UTR region of the *ZFP57* gene is hypermethylated in buccal cells tissue from autistic individuals. (**a**) Upper panel depicts coordinates in chromosome 6 (hg19). (**b**) Represents the corresponding annotation data according to BioMart database. (**c**) Methylation data are shown as beta values, smoothed lines denote mean methylation levels for female ASD (red), male ASD (blue) and typically developing males (purple) and females (green). Each point represents the methylation level of a particular individual at a specific genomic location. (**d**) Copy number estimate obtained by calculating the difference between total probe intensity for a given individual and the mean total intensity across all individuals for a given probe. ASD subjects in orange, TD in gray.

**Figure 3 children-09-00462-f003:**
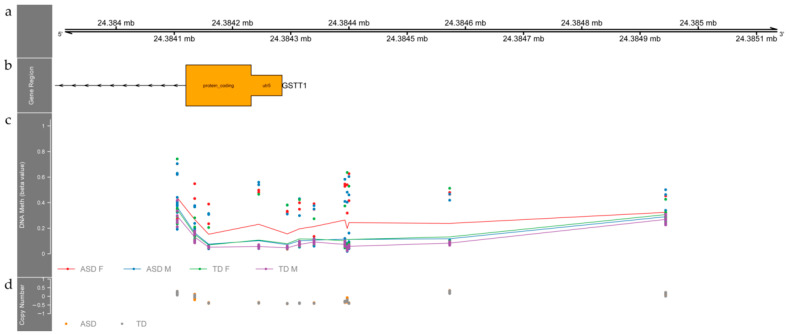
A genome-wide differentially methylated region (DMR) located in the 5′UTR region of the *GSTT1* gene is hypermethylated in buccal cell tissue from autistic individuals. (**a**) Upper panel depicts coordinates in chromosome 12 (hg19). (**b**) Represents the corresponding annotation data according to the BioMart database. (**c**) Methylation data are shown as beta values, smoothed lines denote mean methylation levels for female ASD (red), male ASD (blue) and typically developing males (purple) and females (green). Each point represents the methylation level of a particular individual at a specific genomic location. (**d**) Copy number estimate obtained by calculating the difference between total probe intensity for a given individual and the mean total intensity across all individuals for a given probe. ASD subjects in orange, TD in gray.

**Table 1 children-09-00462-t001:** Age and gender characteristics of ASD and TD groups.

	ASD	TD	*p*-Value
Subject Age (mean, SD)	5.2, 1.9	5.6, 1.09	0.7645 ^1^
(range)	3–12	4–7	
Gender			
Male:Female	22:5	6:9	0.01479 ^1^

Subject age is presented as mean age and standard deviation (SD). Number of subjects of each gender per group. ^1^
*p*-value test is described in methods. Abbreviations: Autism Spectrum Disorder (ASD); Typically developing subjects (TD).

**Table 2 children-09-00462-t002:** Differentially methylated regions between individuals with ASD and TD individuals.

Gene ^a^	Chromosome	Width (Base Pairs)	False Discovery Rate Corrected *p*-Value	Methylation Differences (ASD-TD) ^b^
*FAIM*, Fas Apoptotic Inhibitory Molecule 2	chr12	469	3.18 × 10^−3^	−0.087
*CPXM2*, Carboxypeptidase X, M14 Family Member 2	chr10	674	4.20 × 10^−3^	0.056
*NRIP2*, Nuclear Receptor Interacting Protein 2	chr12	592	8.99 × 10^−3^	0.052
*SOX7*, *CTD-2135J3.3* SRY-Box7	chr8	205	3.03 × 10^−2^	−0.061

^a^ Location within gene sequence or nearest gene. ^b^ Methylation differences are defined as mean beta values of ASD minus mean beta values of TD individuals.

## Data Availability

Data available on request due to restrictions, e.g., privacy or ethical. The data presented in this study are available on request from the corresponding author. The data are not publicly available due to privacy reasons.

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
