# Peer review of "Epigenome-Wide Analysis Reveals DNA Methylation Alteration in ZFP57 and Its Target RASGFR2 in a Mexican Population Cohort with Autism"

_children, 2022, doi:10.3390/children9040462_

Round 1

Reviewer 1 Report

In this work, Aspra at al performed a genome-wide methylation study on buccal epithelium cells from autism spectrum disorder (ASD) patients and typically developing controls. As the authors underline, the analysis of DNA methylation in peripheral tissues from patients with ASD is precious due to the unavailability of brain tissue. Overall, the results are interesting. I have some comments and suggestions for the authors.

- I recommend a careful English editing, revising subject-verb agreement (e.g., “autism spectrum disorders (ASD) comprises…”).

- Some gene names are not italicized. Please follow the guidelines for formatting gene and protein names throughout the manuscript (symbols for genes are italicized, whereas symbols for proteins are not italicized). 

- In the title and along the manuscript, I would suggest indicating “DNA methylation alteration” instead of “differential DNA methylation” since the term “differential” implies a comparison.

- Introduction

I suggest to cite at least two reviews (doi:10.3390/ijms21218290 and https://doi.org/10.1186/s13229-020-00370-1) in support to this sentence “Several factors contributing to ASD etiology have been identified, including genetic and environmental factors.”

Please, in the context of this sentence “Regarding genetic factors, there is strong evidence of their key role in the vulnerability to ASD”, highlight that the main genetic alterations in ASD are involved in specific pathways such as synaptic function (for example, these works can be cited: https://doi.org/10.1016/j.neuron.2011.05.021; doi:10.3390/jcm8020212; https://doi.org/10.3389/fncel.2018.00470; doi: 10.1101/cshperspect.a009886)  

Moreover, I suggest to comment and cite the work of Smith et al. (doi:10.1002/ajmg.b.32278) that compared saliva and blood methylomes with methylation patterns in different brain tissues, finding an higher similarity between salivary methylome and methylation patterns in each of the brain regions analysed than methylation in blood. It can be cited in the Introduction and also in the discussion when it is highlighted that buccal epithelial and brain cells are derived from the ectodermal layer during development and thus DNA methylation in saliva may be more consistent with methylation patterns in brain.

Related to these sentences “Also, DNA methylation differences were evaluated in buccal epithelial cells from individuals with ASD born to mothers of > 35 years. This study detected 13 differentially methylated regions (DMRs) mainly over genes with copy number variants (CNV) reports”, it should be indicated that the DMRs were identified in an analysis of ASD and typically developing controls.

- Methods

It is not clear in which analysis, the authors “considered as significant a nominal P value < 0.1”.

Please justify the importance of the identification of differentially methylated regions between ASD and TD children considering sex as an interactive variable.

- Results

In the paragraph 3.2, I found a discrepancy between Methods and the FDR reported here (FDR < 0.005), please check.

In the paragraph 3.3 “Methylation differences (β, ASD-TD) < 0.1”, I think that the sign is not correct, since the authors selected methylation differences greater than 10%, please revise also considering the detection of both hypermethylated and hypomethylated regions.

I think that “with subjects” should be changed in “with ASD” in this sentence “On average, females and males with subjects”.

I cannot understand why RASGRF2 was not identified in the previous analysis, and it is not clear where these values “(p value= 0.00033, mean beta fold change= -4.036155e-02, mean FDR corrected p-value = 2.156111e-06)” come from.

I believe it would be interesting to add the localization of CpG islands and or/shore and shelf, represented as boxes, in the Figures.

- Discussion

In paragraph 4.2, the identified differentially methylated region is at 5’ of ZFP57. Please check the following sentence “the region we identified extends 200 base pairs downstream the coding region of ZFP57”.

Please describe with more details the association between ZFP57 and RASGRF2.

Please revise this sentence “Therefore, future research with larger samples and individuals of both sexes is necessary to confirm our results and their association with suicide.”

Reviewer 2 Report

please see attached.
